# Vps3 and Vps8 control integrin trafficking from early to recycling endosomes and regulate integrin-dependent functions

Caspar T.H. Jonker [1,5], Romain Galmes [1,6], Tineke Veenendaal[1], Corlinda ten Brink[1], Reini E.N. van der Welle[1], Nalan Liv[1], Johan de Rooij [2], Andrew A. Peden [3], Peter van der Sluijs[1,7], Coert Margadant[4] & Judith Klumperman [1]

Recycling endosomes maintain plasma membrane homeostasis and are important for cell polarity, migration, and cytokinesis. Yet, the molecular machineries that drive endocytic recycling remain largely unclear. The CORVET complex is a multi-subunit tether required for fusion between early endosomes. Here we show that the CORVET-specific subunits Vps3 and Vps8 also regulate vesicular transport from early to recycling endosomes. Vps3 and Vps8 localise to Rab4-positive recycling vesicles and co-localise with the CHEVI complex on Rab11-positive recycling endosomes. Depletion of Vps3 or Vps8 does not affect transferrin recycling, but delays the delivery of internalised integrins to recycling endosomes and their subsequent return to the plasma membrane. Consequently, Vps3/8 depletion results in defects in integrin-dependent cell adhesion and spreading, focal adhesion formation, and cell migration. These data reveal a role for Vps3 and Vps8 in a specialised recycling pathway important for integrin trafficking.

[1] Section Cell Biology, Center for Molecular Medicine, University Medical Center Utrecht, Utrecht University, Heidelberglaan 100, 3584 CX Utrecht, The Netherlands. [2] Section Molecular Cancer Research, Center for Molecular Medicine, University Medical Center Utrecht, Utrecht Universty, Heidelberglaan 100, 3584 CX Utrecht, The Netherlands. [3] Department of Biomedical Science, The University of Sheffield, Sheffield, S10 2TN, UK. [4] Department of Molecular Cell Biology, Sanquin Research, Plesmanlaan 125, 1066 CX Amsterdam, The Netherlands. [5]Present address: Department of Ophthalmology, Weill Cornell Medicine, 1300 York Ave, New York, NY 10065, USA. [6]Present address: UCL Cancer Institute, University College London, 72 Huntley Street, London, WC1E 6DD, UK. [7]Present address: Cellular Protein Chemistry, Bijvoet Center for Biomolecular Research, Utrecht University, 3584 CH Utrecht, The Netherlands. Caspar T.H. Jonker and Romain Galmes contributed equally to this work. Coert Margadant and Judith Klumperman jointly supervised this work. Correspondence and requests for materials should be addressed to J.K. (email: j.klumperman@umcutrecht.nl)

The endolysosomal system is important for a variety of cellular processes, such as protein homeostasis, antigen presentation, signal transduction and cell migration. Hence, disruption of endolysosome function is found in a wide range of diseases, from genetic lysosomal storage disorders to cancer and neurodegenerative disorders[1–3]. The progression of cargo through the endolysosomal system, from early endosomes (EEs) to late endosomes and lysosomes or from EEs to the plasma membrane or recycling endosomes (REs), is tightly controlled by dedicated protein machinery. Membrane fusion is coordinated by the concerted action of Rab GTPases, tethers and soluble NSF attachment protein receptors (SNAREs)[4,5]. Rab GTPases drive the process by recruiting effector machinery proteins to specific membrane domains[6,7]. Contact between opposing membranes is then initiated by tethering proteins, followed by SNARE-mediated fusion.

EE-EE fusion is initiated by activation of Rab5, which recruits multiple effector proteins, including the class C core vacuole/endosome tethering (CORVET) complex[8–11]. The hexameric CORVET complex consists of a core (Vps11, Vps16, Vps18 and Vps33A), which is shared with the late endosomal homotypic fusion and protein sorting (HOPS) tethering complex, and additionally contains the two CORVET-specific subunits Vps8 and Vps3 (also named TGFBRAP1 or TRAP1)[9,12]. Recycling of endocytosed proteins and membranes from EEs is crucial to maintain plasma membrane homeostasis and is essential for cell polarity, cell migration and cytokinesis. Recycling occurs either directly from EEs to the plasma membrane (fast recycling) or indirectly via Rab11-positive REs (slow recycling)[13–15]. Both pathways involve Rab4, which resides on EEs as well as on recycling vesicles that emerge from EEs[15–17]. The class C homologues in endosome-vesicle interaction (CHEVI) complex consisting of Vps33B and VIPAS39, homologues of Vps33A and Vps16, respectively, binds to Rab11 and localises to REs[18–20]. Mutations in Vps33B or VIPAS39 underlie arthrogryposis, renal dysfunction and cholestasis (ARC) syndrome, a rare autosomal recessive multisystem disorder that affects the transport of apical and junctional proteins in polarised cells[18,19,21].

A poorly understood step in endosomal recycling is the transport from EEs to REs. Since EEs are the major source of membranes for REs, we here study a possible role for the CORVET complex in endosomal recycling. To our surprise we found that the CORVET-specific Vps3 and Vps8 subunits interact directly with each other and localise to Rab4-positive recycling vesicles and CHEVI-positive REs. Moreover, we show that Vps3 and Vps8 function in a specialised pathway required for integrin recycling, and thereby regulates integrin-dependent cell adhesion and migration.

## Results

**Vps3 and Vps8 localise to recycling vesicles**. The mammalian CORVET complex functions as a tether between EEs and is recruited to membranes via the interaction of Vps8 with Rab5[8]. To determine a possible role of the CORVET complex in endosomal recycling, we analysed the localisation of the CORVET-specific subunits Vps3 and Vps8 in ultrastructural detail. We expressed GFP-Vps3 and HA-Vps8 in HeLa cells and performed immuno-electron microscopy (IEM) by immunogold labelling of ultrathin cryosections. Our IEM data confirmed the localisation of Vps3 and Vps8 on EEs (Fig. 1a, top panel), but in addition revealed a substantial labelling on EE-associated tubules and vesicles (Fig. 1a, bottom panel). The Vps3- and Vps8-labelled vesicles had a characteristic dense content, were often found in clusters (Fig. 1a, lower panel, and 1b) and were consistently negative for endocytosed BSA-Au[5], a marker of endocytic but not

recycling vesicles. These morphological characteristics putatively define the Vps3- and Vps8-positive vesicles as recycling vesicles.

To determine whether the Vps3- and Vps8-positive vesicles also contain CORVET core subunits, we expressed GFP-Vps3 together with FLAG-Vps18 for IEM analysis. Immunogold double labelling showed that Vps18 co-localises with Vps3 on EEs, but was absent from Vps3-positive vesicles (Fig. 1b, asterisks), indicating that Vps3 and Vps8 localise to recycling vesicles, which lack CORVET core components. We then co-expressed GFP-Vps3, HA-Vps8 or the CORVET core subunit GFP-Vps11, together with mCherry-tagged Rab4, Rab5 or Rab11, and assessed their co-localisation by quantitative immunofluorescence (IF) microscopy. We found that Vps3 (Fig. 1d and Supplementary Fig. 1a) and Vps8 (Fig. 1c, d) significantly co-localised with Rab5 and Rab4, which label EEs and EE-derived recycling vesicles, respectively[17,22], and partially with the RE marker Rab11 (Fig. 1c, d and Supplementary Fig. 1a), indicative for the presence of Vps3 and Vps8 on EEs as well as on recycling vesicles. By contrast, GFP-Vps11 only notably overlapped with Rab5 (Fig. 1d and Supplementary Fig. 1b), reinforcing the notion that CORVET core components are present on EEs but not on recycling vesicles. To facilitate the distinction of Rab5 and Rab4 subdomains within EEs by IF microscopy, we expressed the constitutively active Rab5Q79L mutant, which increases the size of EEs[22,23]. Strikingly, co-expression of GFP-Rab5Q79L with HA-Vps8 and FLAG-Rab4 revealed that HA-Vps8 specifically concentrated in FLAG-Rab4-positive patches (Fig. 1e, arrows) and protrusions (Supplementary Fig. 1c, arrows) of GFP-Rab5Q79L-positive EEs. To further confirm that Vps3 and Vps8 reside on the same recycling vesicles as Rab4 we performed IEM of HeLa cells expressing HA-Vps8 or HA-Vps3 with GFP-Rab4. This unequivocally demonstrated that Vps3 and Vps8 indeed co-localise with Rab4 on recycling tubules and vesicles (Fig. 1f, arrows).

In conclusion, these data reveal that Vps3 and Vps8 are present on EE-derived Rab4-positive recycling vesicles and Rab11-positive REs, which lack CORVET core components.

**Vps3 and Vps8 interact directly with each other**. Previous studies showed that Vps3 and Vps8 bind the CORVET core complex[9,12]. In addition, direct interactions between Vps3 and Vps8 have been reported in yeast[24]. Since we find that Vps3 and Vps8 co-localise on recycling vesicles that lack CORVET core components, we performed co-immunoprecipitation (co-IP) and glutathione S-transferase (GST)-binding experiments to investigate whether mammalian Vps3 and Vps8 bind directly to each other. In agreement with this scenario, co-IP experiments in HeLa cells revealed that endogenous Vps8 readily co-immunoprecipitated with overexpressed GFP-Vps3, while less Vps8 was recovered with GFP-tagged Vps11, Vps16 or Vps41, the HOPS-specific subunit that is most homologous to Vps8 (Fig. 2a, full blots in Supplementary Fig. 2). To investigate whether Vps3 can directly bind Vps8 we probed the binding of full-length GST-Vps3 purified from bacterial lysates with in vitro-translated HA-Vps8. Importantly, GST-Vps3 interacted directly with HA-Vps8 (Fig. 2b, full blots in Supplementary Fig. 3). By expressing truncation mutants of GST-Vps3, we could map the Vps8 interacting domain to the C terminus of Vps3 (Fig. 2b), which also confirms the specificity of the binding assays. Finally, to quantify the interaction of Vps3 with Vps8, Vps11 or Vps16 we performed a proximity ligation assay (PLA) in HeLa cells. This showed a significantly higher PLA signal between Vps3 and Vps8, than between Vps3 and Vps11 or Vps16 (Fig. 2c, d).

Taken together, these data show that Vps3 and Vps8 can interact directly with each other.

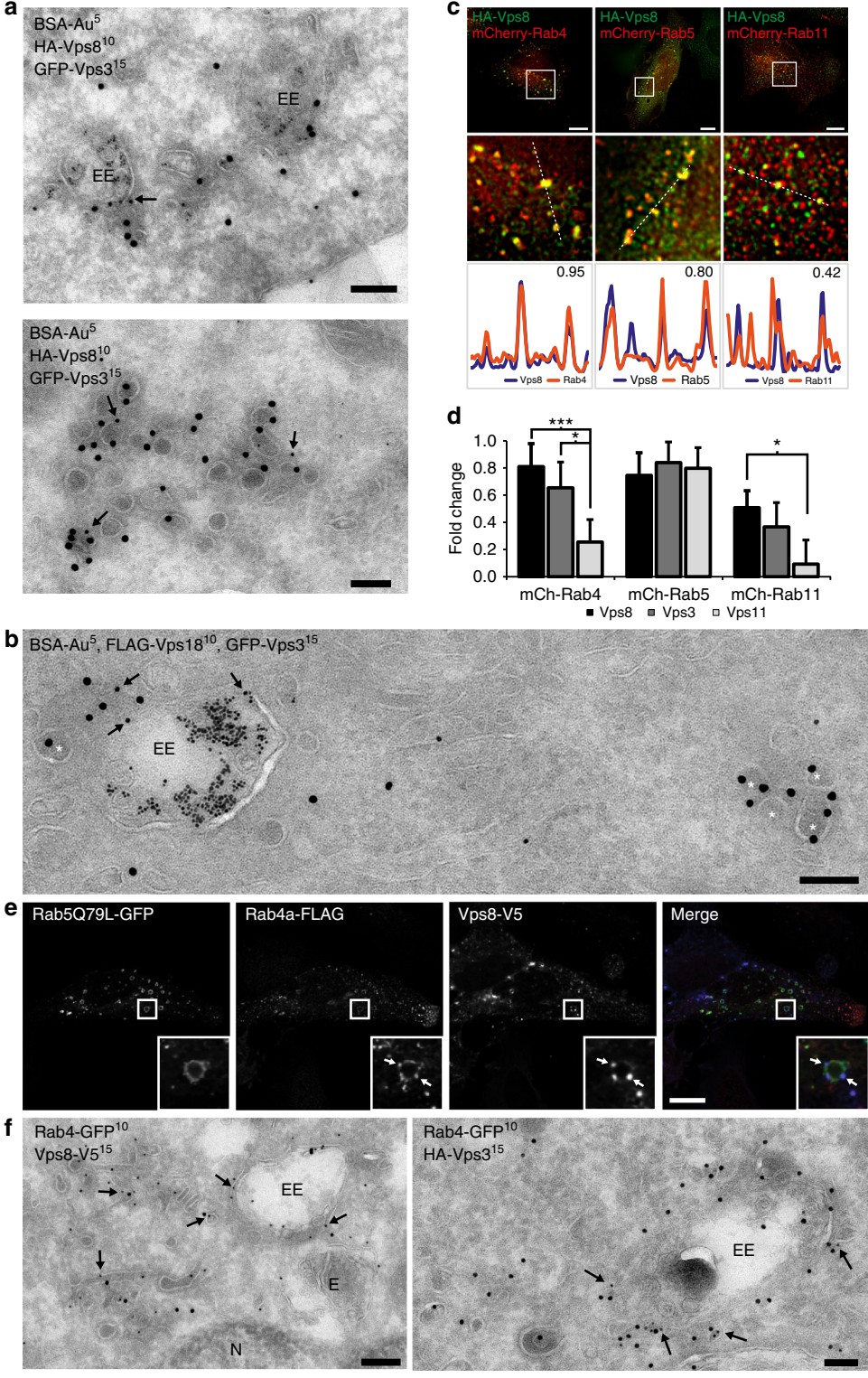

**Fig. 1** Vps3 and Vps8 localise to Rab4- and Rab11-positive recycling endosomes. **a** IEM of HeLa cells expressing HA-Vps8 and GFP-Vps3, and loaded with BSA-Au5. HA-Vps8 (10 nm gold, arrows) and GFP-Vps3 (15 nm gold) co-localise on EEs (top panel) and on vesicles negative for BSA-Au5 (bottom panel). Bar, 100 nm. **b** IEM of HeLa cells expressing FLAG-Vps18, GFP-Vps3 and HA-Vps8, and loaded with BSA-Au5. FLAG-Vps18 (10 nm gold, arrows) and GFP-Vps3 (15 nm gold) co-localise on EEs but not on vesicles negative for BSA-Au5 (asterisk). Bar, 100 nm. **c** HeLa cells expressing HA-Vps8 together with mCherry-Rab4, mCherry-Rab5 or mCherry-Rab11. HA-Vps8 co-localises strongly with Rab4 and Rab5, and less with Rab11. Bar, 10 μm. **d** Quantification of triplicate experiments shown in **c** based on $n > 30$ cells per condition. Error bars represent the standard error of the correlation coefficient (SEr). *$P \leq 0.05$, ***$P \leq 0.001$, calculated using Eqs. 4 and 5. **e** IF of HeLa cells expressing Rab5Q79L-GFP, Rab4-FLAG and Vps8-V5. Vps8 localises to Rab4-positive patches on Rab5Q79L-positive enlarged endosomes (arrows). Bar, 10 μM. **f** IEM of HeLa cells expressing Rab4-GFP together with Vps8-V5 or HA-Vps3, showing co-localisation of Rab4-GFP (10 nm gold, arrows) with Vps8-V5 (15 nm gold, left panel) and HA-Vps3 (15 nm gold, right panel) on recycling tubules and vesicles. Bar, 200 nm. E endosome, EE early endosome, N Nucleus. *$P \leq 0.05$, ***$P \leq 0.001$ using Student's $t$-test

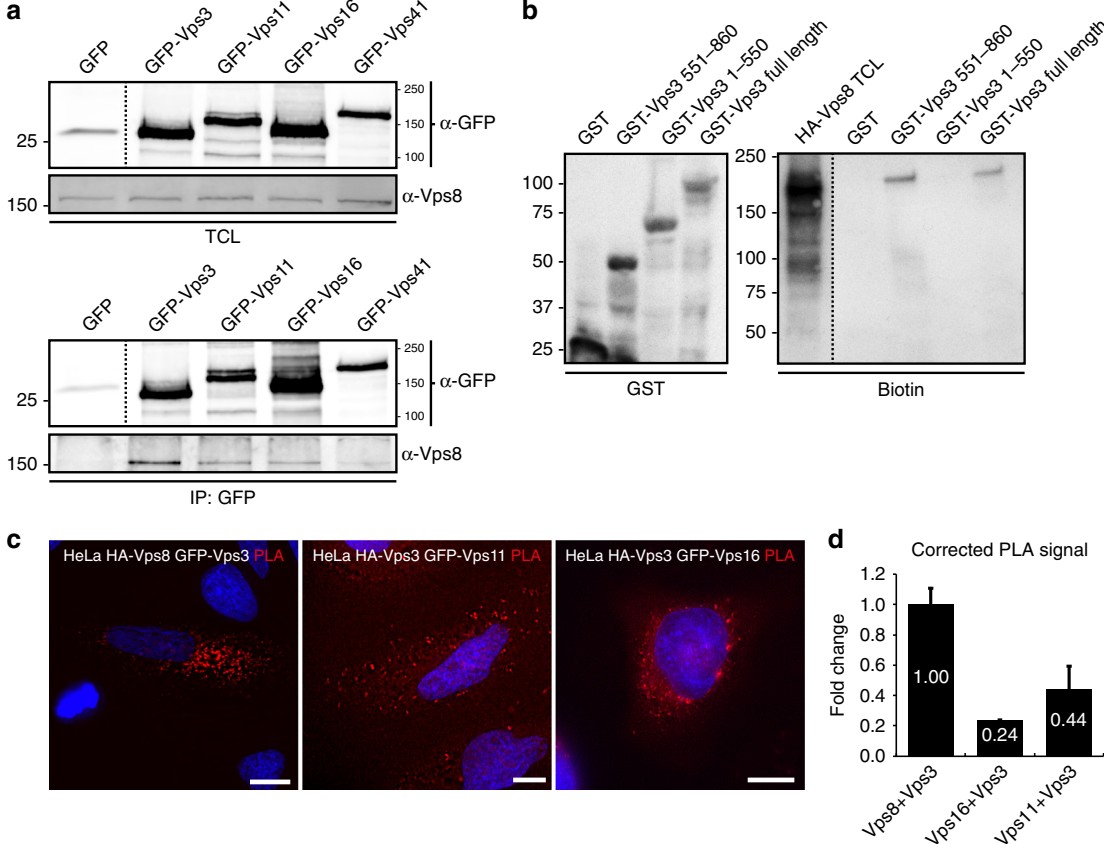

**Fig. 2** Vps3 and Vps8 interact with each other. **a** IPs of GFP-tagged Vps3, Vps11, Vps16 or Vps41 probed for interaction with endogenous Vps8 (GFP control lane shows area around 25 kDa from the same membrane). **b** IPs of GST-tagged Vps3 and truncation mutants of GST-Vps3 purified from bacterial lysates (left panel) probed for interaction with in vitro-translated HA-Vps8. In vitro translation was performed with biotinylated lysines using the Transcend Non-Radioactive Translation Detection System (Promega) according to the manufacturer's protocol and detected using streptavidin-HRP. HA-Vps8 binds to the C terminus of Vps3 and full-length Vps3. **c** PLA on HeLa cells. The highest PLA signal is between Vps3 and Vps8. Bar, 10 μm. **d** Quantification of triplicate experiments shown in **c** based on *n* > 30 cells per condition, PLA signal corrected for protein expression and cell size. Error bars represent standard deviation of the mean (SD)

**Vps3 and Vps8 co-localise with CHEVI on recycling membranes**. The CHEVI complex interacts with Rab11 and localises to Rab11-positive REs[8,9,18,19,25,26]. Since Vps3 and Vps8 partially co-localise with Rab11 (Fig. 1c, d) we investigated co-localisation between Vps3 and Vps8 with the CHEVI complex. To allow formation of the CHEVI complex we co-expressed VIPAS39-mCherry and untagged Vps33B. Expression of the CHEVI complex with either HA-Vps8 or GFP-Vps3 resulted in clear co-localisation as determined by quantitative IF microscopy (Fig. 3a, b). By IEM, these foci of co-localisation were identified as Vps3-, Vps8- and CHEVI-positive clusters of vesicles and tubules, which is the typical morphology for REs[27] (Fig. 3c).

To examine possible interactions between the CHEVI complex and Vps3 or Vps8, or between CHEVI and the CORVET core subunits, we performed co-IP experiments after co-expression of HA-VIPAS39 with GFP-tagged constructs of Vps3, Vps8, Vps11, Vps16, Vps33A, Vps33B or Vps41. Because co-expression of Vps33B together with VIPAS39 is needed for membrane recruitment of the CHEVI complex[18,19], we included an internal control by probing the interactions with HA-VIPAS39 in the absence (Fig. 3d, left panel) or presence (Fig. 3d, right panel) of overexpressed HA-Vps33B. As expected, we detected a strong interaction between HA-VIPAS39 and GFP-Vps33B (Fig. 3d, lane 7 both panels, full blots in Supplementary Fig. 4). Strikingly, however, no significant binding between CHEVI with any of the other subunits was found (Fig. 3d). Only after prolonged

exposure of the western blot a weak interaction between GFP-Vps3 and HA-VIPAS39 was detected (Supplementary Fig. 5 and 6). Previous studies using endogenous proteins failed to detect specific interactions between VIPAS39 with any of the HOPS or CORVET subunits, including Vps3 and Vps8[8,9]. Our studies are in line with these findings, however, the weak binding between CHEVI and Vps3 could be indicative for transient interactions.

Taken together, our data show that in addition to their localisation on Rab4-positive recycling vesicles (Fig. 1f), Vps3 and Vps8 partially co-localise with the CHEVI complex on REs. Furthermore, these data suggest that a transient interaction of Vps3 and Vps8 with the CHEVI complex may occur on REs.

**Vps3 and Vps8 are required for recycling of β1 integrins**. To assess which cargo is transported by the Vps3/8-positive recycling vesicles, we first determined the effect of Vps3 and Vps8 depletion on transferrin (Tf) recycling. We depleted both subunits to fully prevent any function related to the interaction between residual Vps3 and Vps8. Endocytosed Tf recycles either directly from EEs to the plasma membrane or indirectly via REs, while dextran is transported to the lysosome for degradation[28]. A block in recycling increases inclusion of Tf in the degradative pathway to the lysosome, and therefore increases the co-localisation between Tf and dextran[29,30]. Vps3/8 depletion had no effect on Tf localisation, indicating that the recycling of Tf was not disturbed

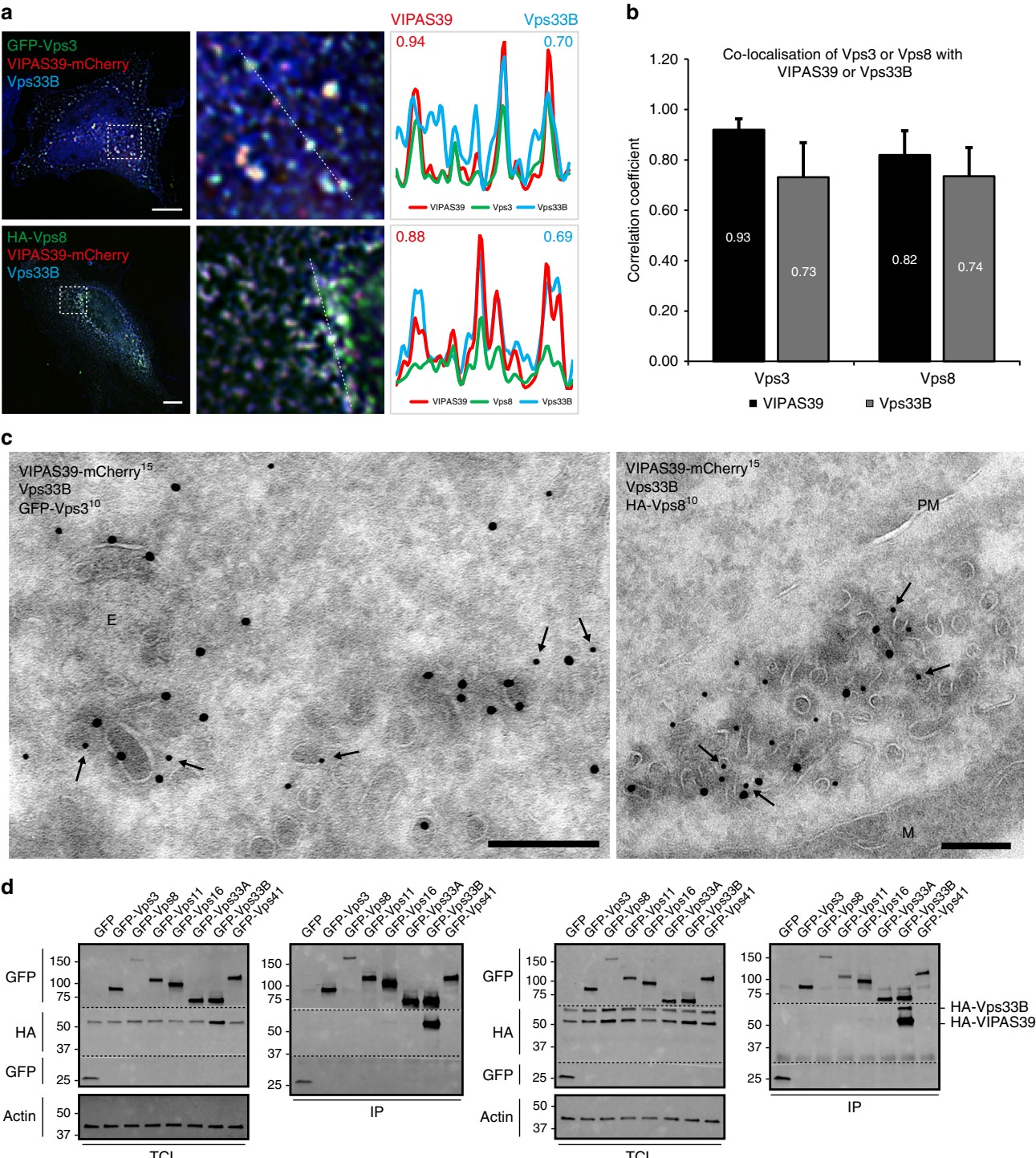

**Fig. 3** Vps3 and Vps8 co-localise with the CHEVI complex. **a** IF of HeLa cells expressing VIPAS39-mCherry, untagged Vps33B and either GFP-Vps3 or HA-Vps8. Vps3 and Vps8 co-localise with both VIPAS39 and Vps33B (line profiles in right panel indicate co-localisation correlation, correlation coefficients indicated: VIPAS39 = red; Vps33B = blue). Bar, 10 μm. **b** Quantification of triplicate experiments shown in **a** based on $n > 30$ cells per condition by line profile correlations. The mean is calculated using Fishers' $r$ to $z$ transformation. Error bars represent SEr, calculated using Eqs. 4 and 5. **c** IEM analysis of HeLa cells expressing VIPAS39-mCherry (15 nm gold), untagged Vps33B (not labelled) and either GFP-Vps3 (10 nm gold, arrows) or HA-Vps8 (10 nm gold, arrows). Vps3 (left panel) and Vps8 (right panel) co-localise with VIPAS39 on recycling vesicles and tubules. E endosome, M mitochondria, PM plasma membrane. Bar, 200 nm. **d** IPs of GFP, GFP-tagged Vps3, Vps8, Vps11, Vps16, Vps33A, Vps33B or Vps41 probed for interaction with co-expressed HA-VIPAS39 (left panel) or with co-expressed HA-VIPAS39 and HA-Vps33B (right panel) in HeLa cells. Only Vps33B shows a strong interaction with VIPAS39

(Fig. 4a, b). These data are in line with a previous study showing that Vps3 is not required for Tf recycling[8].

In addition to the common recycling pathway taken by Tf, specific recycling pathways are described for cell-surface proteins such as integrins[31–33]. To study a potential role of Vps3 and Vps8 in the recycling of β1 integrins we used an established antibody-based integrin recycling assay[34]. HeLa cells were depleted of Vps3 and Vps8, and then serum-starved overnight, labelled with anti-β1 integrin, washed and chased for 2 h. In the absence of serum, β1 integrins accumulate in REs (Fig. 4c, arrows). Vps3/8 knockdown prevented the accumulation of β1 integrins, which instead retained a dispersed distribution throughout the cell

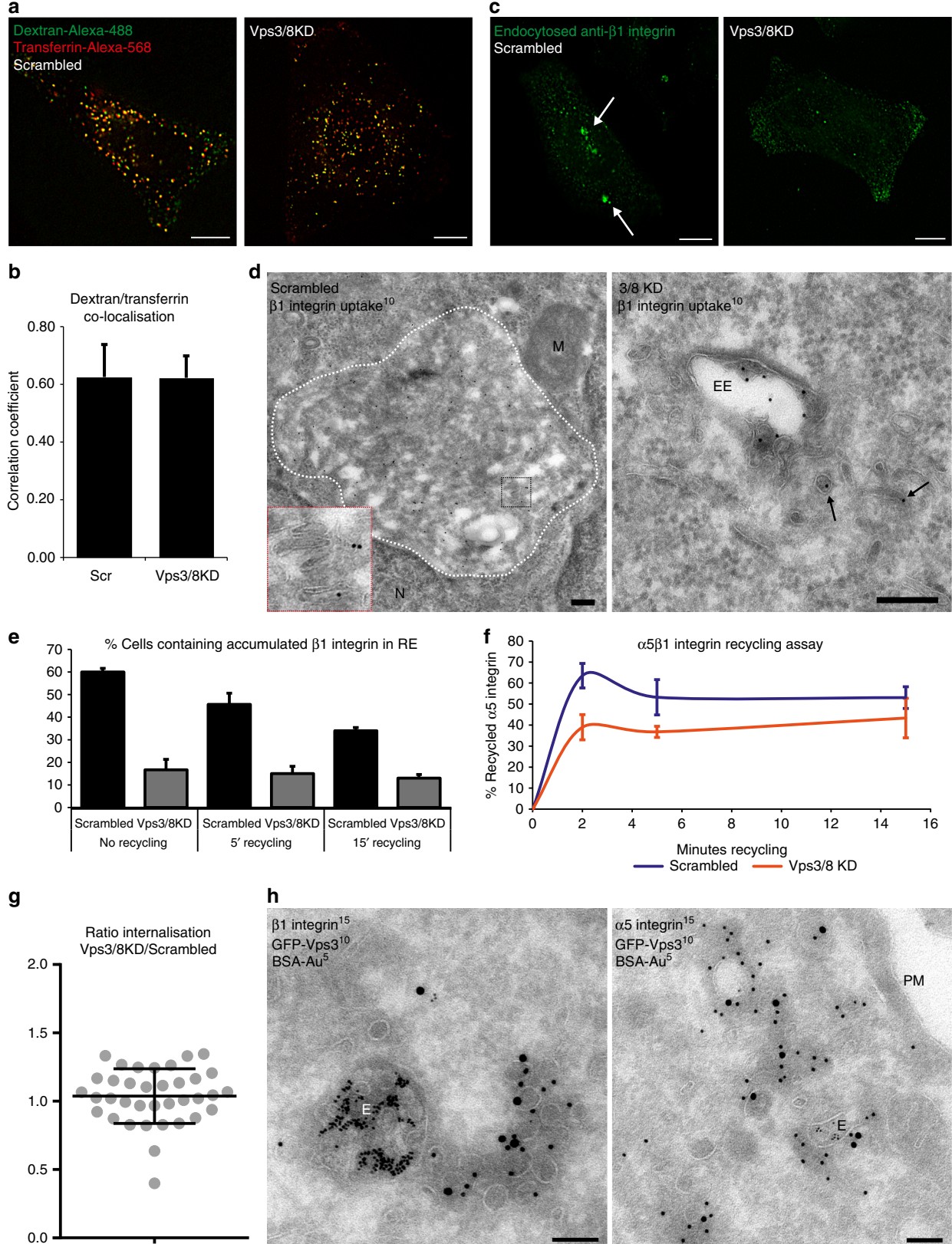

(Fig. 4c and Supplementary Fig. 7). Using an adapted IEM protocol to visualise the antibody uptake as described above, we could identify the accumulations in control cells as large clusters of recycling tubules and vesicles (Fig. 4d, left panel). Importantly, IEM of the Vps3/8 knockdown cells showed that internalised β1 integrins still entered EEs and EE-associated recycling vesicles (Fig. 4d, right panel). These data show that the block in β1 integrin recycling induced by Vps3/8 knockdown occurs after exit from EEs, and interferes with the transport of recycling vesicles to REs. Re-addition of serum induced integrin recycling in control cells resulting in the disappearance of intracellular accumulations, while the localisation of internalised β1 integrins in knockdown cells remained unchanged (Fig. 4e and Supplementary Fig. 7). Together these results imply that Vps3/8 knockdown prevents or delays internalised β1 integrins to reach REs and thereby prevents clearance of the intracellular pool.

We next investigated the impact of Vps3/8 knockdown on the kinetics of integrin recycling by studying the fibronectin (FN)-binding integrin α5β1[35]. In short, cells were surface-labelled with cleavable biotin and endocytosis was allowed for 30 min in the absence of serum. Surface biotin was removed and recycling was induced by re-addition of serum. Surface biotin was removed again and the reduction of intracellular biotinylated α5 integrin before and after recycling was quantified using capture ELISA. This assay showed that knockdown of Vps3/8 resulted in a significant decrease of α5β1 recycling, especially after short recycling times (Fig. 4f), while the internalisation of α5β1 as not affected (Fig. 4g). Finally, to verify that α5β1 is indeed transported by Vps3/8-positive recycling vesicles, we labelled the α5 and β1 integrin subunits for IEM. This unequivocally demonstrated that integrin α5β1 is present in the Vps3/8-positive recycling vesicles (Fig. 4h).

Taken together, these data show that β1 integrins are incorporated into Vps3/8-positive recycling vesicles, and that Vps3 and Vps8 are required for efficient β1 integrin recycling.

**Vps3 and Vps8 regulate cell adhesion and migration.** Integrins are the primary receptors for extracellular matrix (ECM) proteins[36] and integrin recycling is important for cell-ECM interactions and cell migration[37–39]. To investigate whether Vps3 and Vps8 are required for integrin-dependent cell adhesion, we seeded HeLa Vps3/8 knockdown cells on FN or collagen-I (Col-I) in the absence of serum. Non-adherent cells were washed away after 5, 15 or 30 min, and the number of attached cells was quantified. This revealed that adhesion to both FN (Fig. 5a) and Col-I (Fig. 5b) was significantly compromised in Vps3/8 knockdown cells. Individual knockdown of Vps3 or Vps8 also reduced cell

adhesion, but the effect was less stringent, suggesting that residual expression in single-knockdown cells still allows a functional interaction between Vps3 and Vps8, while in the double-knockdown cells this is more completely abrogated (Supplementary Fig. 8a). Importantly, different siRNA oligos induced essentially the same results, rendering it unlikely that the observed adhesion defect is due to off-target effects (Supplementary Fig. 8b).

Because integrin-dependent cell adhesion is followed by a reorganisation of the cytoskeleton and cell spreading, we next determined cell spreading on FN and Col-I 2 h after cell adhesion (Fig. 5c). Quantification of the number of spread cells showed that Vps3/8 depletion reduces cell spreading on both FN and Col-I (Fig. 5d). Integrin-mediated cell spreading is linked to the assembly of focal adhesions (FAs), which anchor the actin cytoskeleton to the ECM[40]. To determine if Vps3/8 knockdown impacts on FA formation, we quantified the number of FAs per cell using confocal microscopy on cell populations stained for vinculin, a well-established marker of FAs (Fig. 5e). Consistent with reduced cell adhesion and spreading, Vps3/8 knockdown cells assembled less FAs, both on FN and Col-I (Fig. 5f).

Finally, we analysed cell motility by time-lapse microscopy (Fig. 5g, arrows indicate migration direction). Knockdown of Vps3 and Vps8 resulted in reduced migration speed (Fig. 5h) and migration distance (Fig. 5i) on both Col-I and FN. Moreover, analysis of cell morphology revealed that the knockdown induced a tail retraction defect (Fig. 5g, j), which is a known effect of impaired integrin trafficking[41,42].

Together these results show that interfering with Vps3/Vps8-dependent integrin recycling leads to a variety of defects in integrin-dependent processes, including cell adhesion and migration.

## Discussion

Previous studies on mammalian Vps3 and Vps8 established their importance in EE-EE fusion as part of the CORVET complex[8]. Here we show that Vps3 and Vps8 independent of CORVET are required for recycling of selected cargo proteins. By IEM and IF, we show that Vps3 and Vps8 co-localise on EEs as well as on Rab4-positive, EE-associated recycling vesicles and tubules that lack CORVET core components. In addition, we find Vps3/8 on Rab11-positive REs. By co-IP and assays using purified proteins we show strong interactions between Vps3 and Vps8, demonstrating that they can interact directly with each other. In addition, we show by IEM that Vps3 and Vps8 co-localise with the CHEVI complex on REs. Based on our data we propose that Vps3 and Vps8 are recruited to EEs together with CORVET core

**Fig. 4** Vps3 and Vps8 are required for recycling of β1 integrins but not Tf. **a** HeLa cells knocked down for Vps3, Vps8 or Vps3/8 were incubated with green fluorescent dextran and red fluorescent Tf for 2 h and visualised by fluorescence microscopy. **b** Quantification of **a**, co-localisation analysis from three separate experiments with n > 40 cells showed no block of Tf recycling. Error bars represent SD. **c** Integrin recycling assay as described[34]. HeLa cells knocked down for Vps3/8 were serum-starved overnight and labelled with anti-β1 integrin on ice. After 2 h uptake at 37 °C, followed by fixation and labelling with fluorescent secondary antibodies, β1 integrin accumulation in REs (arrows) was visualised by IF. In knockdown cells, no accumulation of integrin was found, indicating that the trafficking to REs is blocked. **d** IEM of cells subjected to the integrin uptake assay as in **c** and immunogold-labelled for endocytosed anti-β1 integrin (10 nm gold). Accumulations of endocytosed β1 integrin are by IEM identified as a cluster of tubular membranes (left panel, cluster accented by white dashed line, inset shows magnification of membrane tubules). In knockdown cells, internalised β1 integrin were found in EEs and associated vesicles (right panel, arrows). Bar, 200 nm. **e** Quantification of Supplementary Fig. 7, indicating the percentage of cells with accumulation of β1 integrins in the RE. n = 300 cells per time point from three separate experiments. Error bars represent SD. **f** ELISA-based recycling assay of surface-biotinylated α5 integrin as described[35]. HeLa cells knocked down for Vps3/8 show a decrease in recycling of α5β1 integrin compared to the control. Error bars represent SD. **g** Internalisation of integrins measured using the ELISA data of **f**. The experimental replicates were compared using the ratio between Vps3/8 KD and scrambled cells. The ratio of ~1.0 indicates no significant difference in internalisation between conditions. Error bars represent SD. **h** IEM of HeLa cells loaded with BSA-Au5 expressing GFP-Vps3. GFP-Vps3 (10 nm gold) co-localises with the endogenous β1 (15 nm gold, left panel) and α5 integrin subunits (15 nm gold, right panel). Bar, 100 nm. PM plasma membrane, E endosome, EE early endosome, N nucleus, M mitochondrium

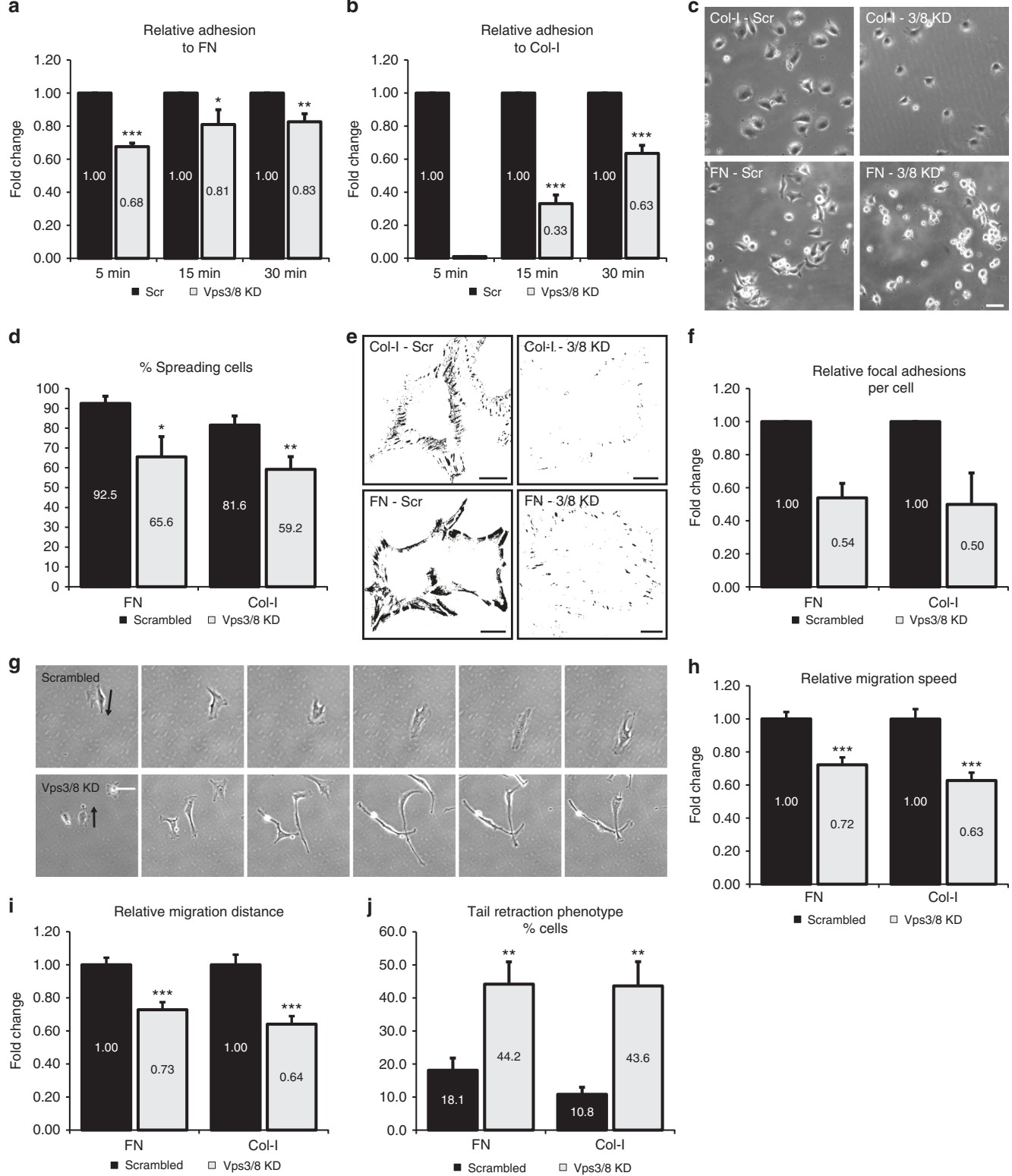

**Fig. 5** Vps3 and Vps8 regulate β1 integrin-dependent cell adhesion and migration. **a**, **b** Cells knocked down for Vps3/8 show a decreased capability to attach to FN (**a**) or Col-I (**b**). Error bars indicate SD. **c** HeLa cells knocked down for Vps3/8 show decreased spreading compared to the control cells 2 h after seeding on Col-I- or FN-coated surfaces. Bar, 100 μm. **d** Quantification of triplicate experiments of **c**. Error bars represent the SD. **e** Background-subtracted and thresholded confocal images of HeLa cells knocked down for Vps3/8 and stained with rabbit anti-vinculin. Knockdown cells show a reduction in the number of FAs per cell both on Col-I and FN. Bar, 10 μm. **f** Quantification of triplicate experiments of **e**. **g** Time-lapse microscopy of HeLa cells knocked down for Vps3/8 (FN condition, arrows indicate direction of migration) show defects in mean migration speed. Bar, 100 μm. Relative migration speed (**h**) and mean migration distance (**i**) on both Col-I and FN are reduced. Error bars represent the standard error of the mean (SEM). **j** The percentage of cells showing an elongated tail during cell migration increases significantly in knockdown cells on both Col-I and FN. Error bars represent SD. ns = $P > 0.05$, *$P \leq 0.05$, **$P \leq 0.01$, ***$P \leq 0.001$ calculated using Student's $t$-test

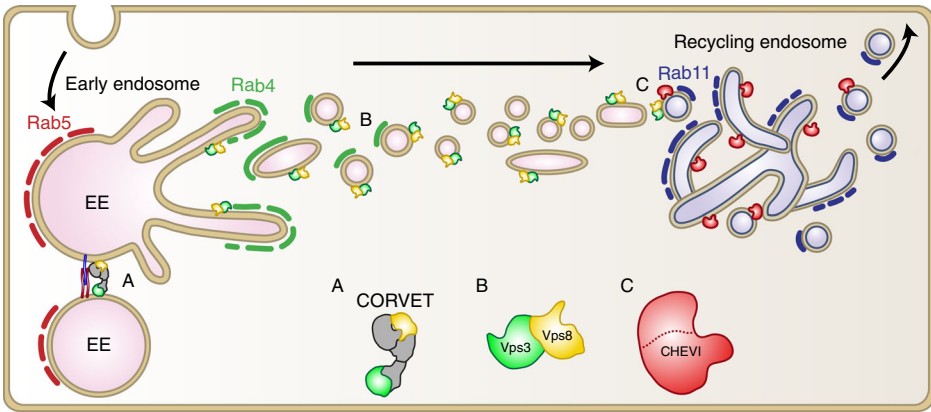

**Fig. 6** Model of the proposed function of Vps3 and Vps8 in endosomal recycling. The CORVET complex mediates the fusion between Rab5-positive EEs (A). Vps3 and Vps8 localise to Rab4-positive vesicles emerging from EEs (B). The Vps3/8-positive recycling vesicles deliver integrins to Rab11- and CHEVI-positive REs (C)

components, after which they localise to Rab4-positive recycling vesicles, which set out to fuse with Rab11- and CHEVI-positive REs (Fig. 6). The Vps3/Vps8-dependent recycling pathway is required for efficient recycling of internalised β1 integrins, and perturbation of Vps3 or Vps8 impairs integrin-dependent cell adhesion and spreading, FA assembly and cell migration.

Rab4 has been implicated in the short-loop recycling from EEs to the plasma membrane, as well as in the long-loop recycling from EEs to Rab11-positive REs[13,15–17,22,43,44]. Since Vps3/8 knockdown prevents delivery of β1 integrins to REs (Fig. 4c, f), we conclude that Vps3 and Vps8 are involved in transport from EEs to REs (Fig. 6). Our finding that Vps3 and Vps8 interact with each other (Fig. 2) and localise to recycling vesicles that lack the CORVET core subunit Vps18 (Fig. 1) could indicate the formation of a Vps3/8 complex independently of the CORVET core subunits. A direct interaction between Vps3 and Vps8 has been found before[8], but seemed counterintuitive since in the current CORVET model Vps3 and Vps8 are located at opposite ends of the complex[45]. The formation of a Vps3/8 complex without other CORVET core components would provide a functional and structural explanation for these findings. Interestingly, Vps39 and Vps41, the counterparts of Vps3 and Vps8 residing on the opposite ends of the HOPS complex, can also directly interact with each other[24], possibly by obtaining a closed post-fusion conformation of HOPS following late endosome-lysosome fusion[45,46]. In analogy herewith, a closed post EE-EE fusion conformation of the CORVET complex could precede the direct interaction of Vps3 and Vps8. Future studies on endogenous proteins are however required to determine if there is indeed a functional Vps3/8 complex formed and whether this requires the presence of additional proteins.

As a crucial mechanism to control integrin function, endocytic recycling of integrins is a topic of extensive research[39,47–49]. Integrins use multiple recycling pathways depending on hetero-dimer composition, conformation (active or inactive) and ligand binding. A key question is how sorting to these different pathways is achieved. Interestingly, we found that Vps3/8 knockdown had no effect on Tf recycling (Fig. 4a, b). Tf recycling depends on Rab4 and Rab11, which are also involved in integrin recycling[13,30,34,38,50]. However, in contrast to Tf, recycling of β1 integrins depends on a NPxY motif that in EEs binds SNX17 or SNX31 to sort integrins into recycling vesicles[51–53]. This suggests that within the Rab4/Rab11 pathway multiple sub-pathways exist. Recently, a new protein complex, retriever, has been identified that associates with SNX17 on EEs and sorts α5β1 integrin from the early endosome to the recycling pathway[54]. Retriever is

therefore a likely candidate to mediate sorting of α5β1 integrin into the Vps3/8 recycling pathway. Together these data suggest that the interplay between CORVET, retriever, CHEVI and a possible Vps3/8 complex is important for integrin recycling. In analogy with the concept of EE-associated tubular sorting endosomes or networks, machinery proteins involved in cargo sorting might define subdomains on REs[27,55,56]. At the level of REs, the CHEVI protein Vps33B is a putative candidate to define exits specialised for integrin recycling, since it interacts directly with integrin β-subunits[57]. Future studies are required to define additional cargo for the retriever-Vps3/8-CHEVI-dependent recycling pathway.

Mutations in the genes that encode for the CHEVI complex cause ARC syndrome, by impairing polarised transport from REs to the apical plasma membrane[18,19,21]. Interestingly, a recently identified patient with a mutation in VPS8 suffered from arthrogryposis[58], one of the hallmarks of ARC patients. Our model that CHEVI and Vps3/8 function in the same pathway provide a possible explanation for this partially overlapping phenotype.

Our data contribute to the emerging view that multi-subunit tethering complexes are dynamic in composition in order to facilitate multiple transport pathways[11,59]. The increasing number of disease-causing mutations found in individual subunits illustrates the importance to understand the role of each individual subunit at a fundamental cellular level.

## Methods

**Cell culture and transfection**. HeLa cells (American Type Culture Collection clone ccl-2) were grown at 37 °C and 5% $CO_2$ in Dulbecco's modified Eagle's medium (DMEM) supplemented with 10% heat-inactivated fetal bovine serum, 2 mM L-glutamine, 100 U/ml penicillin and 100 µg/ml streptomycin. Cells were transfected with cDNA using Effectene transfection reagent (Qiagen) according to the manufacturer's protocol. siRNA transfection was performed using HiPerfect transfection Reagent (Qiagen) according to manufacturer's protocol.

**Antibodies**. For IF and IEM labelling: mouse anti-HA (1:400, 901502, Covance); mouse anti-β1 integrin (1:200, JB1A, Millipore); mouse anti-α5 integrin (1:200, VC5, BD Biosciences); goat anti-HA (1:400, A00168, Genscript); rabbit anti-GFP (TP401, Acris); rabbit anti-mCherry (1:400) was made on site as described[60], and rabbit anti-vinculin was from Abcam (1:500, ab73412). For detection of GST fusion proteins, we used mouse anti-GST (1:5000, sc138, Santa Cruz). Rabbit anti-goat IgG (1:500, 7S, NORDIC) or rabbit anti-mouse IgG (1:250, Z0412, Dako) was used to bridge between goat or mouse antibodies and protein-A gold[61]. Fluorescent secondary antibodies were obtained from Invitrogen (1:250). For immunoprecipitation and western blotting we used mouse anti-GFP (1:2000, 11814460001, Roche), mouse anti-HA (1:2000, 901502, Covance), rabbit anti-mCherry (1:2000) and rabbit anti-GFP (1:2000, ab290, Abcam). Fluorescent secondary antibodies were obtained from Li-Cor (1:20,000).

**Reagents**. Gold particles of 5 nm coupled to bovine serum albumin (BSA-Au[5]), and 10 nm or 15 nm gold particles conjugated to protein A were made on site (Cell Microscopy Core, UMC Utrecht, The Netherlands). HA-Vps8, Vps8-V5 and Vps8-GFP, GFP-Rab4, mCherry-Rab4, FLAG-Rab4, mCherry-Rab5, mCherry-Rab7, mCherry-Rab11 and untagged Vps33B were cloned from cDNA purchased from Origen. HA-Vps3, GFP-Vps3, GFP-Vps11 and Vps41-GFP were a generous gift from Dr. J. Neefjes (NKI, Amsterdam, The Netherlands), VIPAS39-mCherry was a generous gift from Dr. P. Gissen (LMCB, UCL, London), and Vps33B-HA-V5-His was a generous gift from Dr. V. Faundez (Cell Biology, Emory University, Atlanta, GA, USA). Vps8, Vps3, Vps33B and VIPAS39 SMARTPool siRNAs were purchased from Dharmacon. FN and type-I collagen were purchased from Sigma. MG132 was purchased from Enzo Life Sciences. EZ-link Sulfo-NHS-SS-biotin was from Thermo Fisher. Complete protease inhibitors were purchased from Roche. For detection of in vitro-translated proteins, peroxidase-conjugated streptavidin (0.1 μg/ml) was used (Jackson ImmunoResearch).

**GST pulldown**. Full-length Vps3 and Vps3 truncation mutants were cloned in pGEX5 vectors. The proteins were produced in *Escherichia coli* BL21 (DE3) grown at 30 °C and lysed using sonication. Proteins were then immobilised on glutathione sepharose 4B beads (GE). HA-Vps8 was produced using a T7 in vitro translation kit (Promega) in combination with the Transcend Non-Radioactive Translation Detection System (Promega) that incorporates biotin-conjugated methionine. In vitro-translated proteins were incubated o/n at 4 °C with the loaded sepharose 4B Beads. After that beads were washed and proteins were eluted using 1× sample buffer for 30 min at 37 °C. Eluted proteins were separated using SDS-polyacrylamide gel electrophoresis (SDS-PAGE) and analysed using western blotting.

**Co-immunoprecipitation**. Cells were washed in ice-cold phosphate-buffered saline (PBS) and scraped in lysis buffer (40 mM TRIS (pH 7.4), 100 mM NaCl, 0.1% TX-100 and 5 mM EDTA supplemented with complete protease inhibitors (Roche)). Cells were lysed for 30–60 min at 4 °C followed by 20 min centrifugation at 13,000 r.p.m. at 4 °C. Protein A beads were incubated with the desired antibody for 2 h at 4 °C, washed and added to the collected supernatant. Beads and supernatant were incubated in a rotator at 4 °C for 1 h. Beads were washed six times with lysis buffer minus protease inhibitors after which proteins were eluted using 1× sample buffer for 30 min at 37 °C. Eluted proteins were separated using SDS-PAGE and analysed by western blotting.

**Western blotting**. Cells were washed in ice-cold PBS and scraped in E1A lysis buffer (40 mM TRIS (pH 7.4), 100 mM NaCl, 0.1%TX-100 and 5 mM EDTA supplemented with complete protease inhibitors (Roche)). Cells were lysed for 30 min at 4 °C followed by 20 min centrifugation at max speed at 4 °C to remove cell debris. Proteins were eluted using 1.5× sample buffer for 30 min at 37 °C, separated on a 7.5 or 10% SDS-PAGE and transferred to Immobilon-FL PVDF membrane (Millipore). Membranes were blocked at room temperature (RT) using Odyssey blocking buffer in PBS (1:1) for 1 h and incubated with selected primary antibodies diluted in Odyssey blocking buffer in PBS-T 0.2% (1:1). Membranes were washed in PBS-T 0.1% and incubated with secondary antibodies diluted in Odyssey blocking buffer in PBS-T 0.2% (1:1) for 1 h at RT. Finally, membranes were washed and imaged on an Odyssey imaging system (Li-Cor) for co-IP experiments. For GST-pulldown experiments, biotin-labelled proteins were detected using ECL (GE Healthcare), which were consequently exposed to Kodak XB films (Rochester). Complete, uncropped blots are presented in the Supplementary Information.

**IF microscopy**. For Tf recycling experiments, HeLa cells were incubated with dextran conjugated to Alexa-488 (Thermo Fisher) for 120 min and with Tf conjugated to Alexa-568 (Thermo Fisher) for 30 min and imaged in pre-warmed image medium (Sigma) on a climate-controlled DeltaVision widefield microscope using a ×100/1.4 A immersion objective. For fixed-cell IF experiments, HeLa cells grown on sterile glass coverslips were washed with ice-cold PBS and fixed with 4% paraformaldehyde (PFA) in PBS for 20 min at RT. Then, cells were permeabilised using 0.1% TX-100 in PBS for 5 min and blocked for 15 min using PBS supplemented with 1% BSA. Cells were labelled with primary antibodies diluted in blocking buffer at RT for 1 h, washed and labelled with fluorescent secondary antibodies for 30 min in the dark. After labelling, the cells were washed and mounted using Prolong Gold antifade reagent with 4′,6-diamidino-2-phenylindole (DAPI; Thermo Fisher) and imaged on a DeltaVision widefield microscope using a ×100/1.4 A immersion objective. Pictures were deconvolved using Softworx software and analysed using Fiji ImageJ v1.48q.

**Immuno-electron microscopy**. Sample preparation, ultrathin cryosectioning and immunolabelling were performed as described[61]. In brief, HeLa cells were grown on 60 mm dishes and fixed by the addition of freshly prepared 4% PFA in 0.1 M phosphate buffer (pH 7.4) to an equal volume of culture medium for 10 min, followed by postfixation with fresh 4% PFA overnight at 4 °C. Fixed cells were washed in PBS containing 0.05 M glycine and gently scraped in PBS containing 1% gelatine. Cells were pelleted in 12% gelatin in PBS, which was then left to solidify on ice and cut into small blocks. The blocks were infiltrated overnight in 2.3 M

sucrose at 4 °C, mounted on aluminium pins and frozen in liquid nitrogen. The 70 nm ultrathin cryosections were cut on a Leica ultracut UCT cryomicrotome and picked up in a freshly prepared 1:1 mixture of 2.3 M sucrose and 1.8% methylcellulose. Sections were then immunogold-labelled with antibodies and protein A gold, contrasted and examined on a JEOL transmission electron microscope 1010.

For antibody uptake IEM, HeLa cells were serum-starved overnight in DMEM containing 0.01% BSA (D/B) and subsequently incubated at 4 °C for 1 h with anti-β1 integrin (JB1A, Millipore) diluted in D/B to a final concentration of 10 μg/ml. Cells were transferred to pre-warmed D/B, and integrin endocytosis was allowed for 2 h at 37 °C after which cells were prepared for IEM as described above and labelled on section with rabbit-ant-mouse IGG (Z0412, Dako) and protein A 10 nm gold.

**Stimulation-induced integrin recycling assay**. Integrin recycling was analysed essentially as described[34]. In brief, HeLa cells were serum-starved overnight in DMEM containing 0.01% BSA (D/B) and subsequently incubated at 4 °C for 1 h with anti-β1 integrin (JB1A, Millipore) diluted in D/B to a final concentration of 10 μg/ml. Excess antibody was removed by two washes with cold D/B. Cells were then transferred to pre-warmed D/B, and integrin endocytosis was allowed for 2 h at 37 °C. Surface-bound antibodies that were not endocytosed were removed with two acid washes (0.5% acetic acid and 0.5 M NaCl) at 4 °C. For recycling, cells were stimulated with pre-warmed D/B containing 20% fetal calf serum (FCS). The pool of internal β1 integrins was monitored at the indicated time points by fixation and preparation for IF microscopy as described above.

**Cell adhesion and migration assays**. For cell adhesion assays, 96-well plates were coated with 5 μg/ml FN (90 min) or 3 μg/ml Col-I (5 min) at 37 C, washed with PBS and blocked with 2% BSA for 30 min at 37 °C. Cells were trypsinised, resuspended in serum-free DMEM and seeded at a density of $5 \times 10^4$/well. At the indicated time points, non-adherent cells were washed away with PBS. Attached cells were fixed with 4% PFA for 10 min, washed twice with PBS and stained with Crystal Violet (5 mg/ml in 2% ethanol) for 10 min. Stained cells were extensively washed with water and lysed with 2% SDS for 30 min. Plates were read at 590 nm.

For cell-spreading assays, cells were seeded in plates coated with 5 μg/ml FN or 3 μg/ml Col-I, fixed at the indicated time points, and the number of spread cells was counted. For FA analysis, cells were seeded onto coverslips coated with 5 μg/ml FN or 3 μg/ml Col-I, and fixed at the indicated time points. Confocal images of vinculin staining were acquired using fixed settings, background-subtracted and thresholded in ImageJ 1.44, and FA number was then determined using the 'analyse particles' function.

For cell migration assays, cells were sparsely seeded on the indicated matrix proteins, and phase-contrast images were captured every 10 min at 37 °C and 5% $CO_2$ on a widefield CCD system using a ×10 dry lens objective (Carl Zeiss MicroImaging). Migration tracks were generated using ImageJ 1.44, and the average displacement and migration speed were calculated from ~30 cells out of three independent experiments. The tail retraction phenotype was scored from the same movies.

**Proximity ligation assay**. HeLa cells grown on sterile glass coverslips were washed with ice-cold PBS and fixed with 4% PFA in PBS for 20 min at RT. Then, cells were permeabilised using 0.1% Triton X-100 in PBS for 5 min and blocked for 15 min using PBS supplemented with 1% BSA. Cells were labelled with primary antibodies diluted in blocking buffer at RT for 1 h, washed and labelled with mouse PLUS and Rabbit MINUS antibodies for 1 h at 37 °C. The PLA signal was visualised using the in situ PLA detection kit (Sigma) according to the manufacturer's protocol. After labelling the cells were mounted using Prolong Gold antifade reagent with DAPI and imaged on a DeltaVision widefield microscope using a ×100/1.4 A immersion objective. Pictures were deconvolved using Softworx software and analysed using Fiji ImageJ v1.48q.

**Integrin recycling and capture ELISA**. Integrin recycling was investigated essentially as described earlier[35]. Briefly, cells were transferred to ice, washed twice in PBS and surface-labelled at 4 °C with 140 μg/ml NHS-SS-biotin. Internalisation was allowed for 15 min, after which the cells were washed with PBS, and remaining cell-surface biotin was removed with 20 mM MesNa. Recycling of internalised proteins was then induced in DMEM/FCS at 37 °C, whereafter biotin was removed from recycled proteins by a second reduction with MesNa, which was quenched with 20 mM iodoacetamide. Cells were then washed twice in PBS and lysed in 1.5 NDLB buffer (150 mM NaCl, 50 mM Tris, 10 mM NaF, 1.5 mM Na3VO3, 5 mM EDTA, 5 mM EGTA, 1.5% TX-100, 0.75% Igepal CA-630 and 1 mM 4-(2-aminoethyl)benzynesulphonyl fluoride supplemented with complete protease inhibitors). Maxisorb 96-well plates (Life Technologies) were coated overnight with 5 μg/ml anti-integrin α5 in 0.05 M $Na_2CO_3$ (pH 9.6) at 4 °C, and blocked in PBS/0.05% Tween-20 (PBS-T) with 5% BSA. Plates were incubated overnight with cell lysates at 4 °C, washed with PBS-T and incubated with horseradish peroxidase-streptavidin in PBS-T/1% BSA for 1 h at 4 °C. After washing, biotinylated proteins were detected with *ortho*-phenylenediamine in a buffer containing 25.4 mM $Na_2HPO_4$, 12.3 mM citric acid (pH 5.4) and 0.003% $H_2O_2$. The reaction was terminated with 8 M $H_2SO_4$ and absorbance was read at 490 nm.

**Statistical analysis**. For calculation of correlation coefficients of co-localisation in IF experiments the signal intensity over a vector was plotted using the plot profile tool in FIJI ImageJ in the green, red and where needed, the far-red channel. Correlation coefficients between two plots were calculated using the function:

$$\text{correl}(x,y) = \frac{\sum\sum(x-\bar{x})(y-\bar{y})}{\sqrt{(x-\bar{x})^2(y-\bar{y})^2}} \tag{1}$$

Correlation coefficients were calculated between the channels from at least 30 cells taken from three separate experiments. Correlation coefficients were averaged as follows. Individual correlation coefficients ($r$) underwent Fisher $r$-to-$z$ transformation according to the formula:

$$z_r = \frac{1}{2}\ln\left(\frac{1+r}{1-r}\right) \tag{2}$$

The mean of the $z$ values was calculated and the resulting mean was transformed back to the mean correlation coefficient using:

$$r = \frac{e^{2z}-1}{e^{2z}+1} \tag{3}$$

To assess the significance between two mean correlation coefficients the $z$-score and $P$-value were calculated using:

$$z = \frac{\left(\frac{1}{2}\ln\frac{1+r1}{1-r1}\right) + \left(\frac{1}{2}\ln\frac{1+r2}{1-r2}\right)}{\sqrt{\frac{1}{n_1-3} + \frac{1}{n_2-3}}} \tag{4}$$

$$P(Z \le z) = \int_{-\infty}^{z} \frac{1}{\sqrt{2\pi}} e^{\frac{-\mu^2}{2}} d\mu \tag{5}$$

**Data availability**. The data that support the findings of this study are available from the corresponding author upon reasonable request.

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

## Acknowledgements

We thank Adam Grieve and Catherine Rabouille for fruitful discussions and reagents for the initial integrin experiments. Our colleagues of the Center for Molecular Medicine and section Cell Biology for precious input and especially René Scriwanek for preparation of the electron micrographs. Paul Gissen, Victor Faundez and Sjaak Neefjes are kindly acknowledged for sharing critical reagents. Coert Margadant was supported by a grant from the Netherlands organisation for Scientific Research (ZonMW Veni 016.146.160)

## Author contributions

C.T.H.J. designed and performed the experiments needed for Figs. 1–6, analysed the data and wrote the paper. R.G. designed and performed the experiments needed for Figs. 1 and 3, analysed the data and wrote the paper. T.V. performed experiments needed for Figs. 1a, d, 2d, and 4c, f. C.t.B. performed experiments needed for Figs. 2b and 4a and cloned constructs used in the manuscript. R.E.N.v.d.W. performed experiments needed for Figs. 2b and 4e. N.L. performed experiments needed for Fig. 2c, d. J.d.R. designed experiments needed for Fig. 4 and wrote the paper. A.A.P. designed experiments needed for Fig. 3 and wrote the paper. P.v.d.S. designed the experiments for Fig. 1b, c, e, cloned constructs and wrote the paper. C.M. designed and performed the experiments for Figs. 4 and 5, supervised the project, analysed the data and wrote the paper. J.K. initiated and supervised the project, designed experiments for Figs. 1–4 and wrote the paper.

## Additional information

**Competing interests:** The authors declare no competing financial interests.

