## [Peer Review File · Nature Communications]

Reviewers' comments:

Reviewer #1, technical expert in electron microscopy (Remarks to the Author):

All immuno electron microscopic preparations shown in this paper are of excellent qualities and the double-labeling patterns show convincingly what is described in the text. The beautiful results have to be particularly appreciated, since the method of immuno-labelling of ultrathin cryosections is sophisticated, requires special knowledge and skills and the interpretation of the results frequently is a big challenge. This is mainly caused by the fact that in the cryo-sections the ultrastructures may not be such clearly visible as this is the case in sections of resin-embedded materials. In addition, the diagnosis of special ultrastructures and organelles may be difficult, since in ultrathin sections only membrane profiles can be discerned and the actual shapes and architectures of compartments and organelles are not visible.

This leads to my only query concerning the EM part of this manuscript:

In some cases, I would be more cautious with the interpretation of the structures shown. In thin sections, vesicles, tubules and cisternae may appear with the same profiles. In particular, I would like to draw attention to the Golgi apparatus. Recycling endosomes may be located in close vicinity to Golgi apparatus stacks. In some of the images, such as in Fig.1d-top panel and Fig.3e-both panels, structures reminiscent of stacks/mini-stacks of Golgi apparatus cisternae are visible. In the legend to Fig.4f, G=Golgi is indicated but the respective description cannot be found in the picture. Since the spatial vicinity might be of functional significance, it would be of particular interest to show the relation of recycling endosomes to Golgi apparatus compartments.

Reviewer #2, expert in CORVET complex (Remarks to the Author):

In this study, Jonker et al. propose a possible integrin recycling pathway, involving CORVET complex subunits Vps3 and Vps8, as well as the Vps33B/VIPAS39 complex and VAMP3. The authors demonstrate that Vps3 and Vps8 localize to Rab4- and partially on Rab11-positive compartments, whereas the CORVET core does not show any colocalization with these vesicular-tubular structures. Furthermore they show an interaction between Vps3 and Vps8 with the Vps16 homologue VIPAS39, as well as an interplay of Vps33B with the SNARE VAMP3, which is dependent on VIPAS39 co-expression. Upon depletion of either Vps33B or a double knock-down of Vps3 and Vps8, the authors observe a delay in β -Integrin recycling and subsequent problems of cells in adhesion and motility. They propose a model in which Vps3 and Vps8 link Rab4-positive vesicles to Rab11-positive recycling endosomes via a possible interaction between the CORVET subunits Vps3 and Vps8 and the VIPAS39/Vps33B complex. These interactions are supposed to serve in a VAMP3-dependent recycling pathway.

This is an interesting and potentially novel function of two CORVET-specific subunits in context with the dimeric VIPAS-39/Vps33B complex. Their ultrastructural and colocalization analyses, which are certainly of high quality and impressive overall, support this claim, even though colocalization of these subunits does not necessarily indicate interaction outside of the CORVET complex. Likewise, the integrin-specific phenotype seems to be linked to Vps3 and Vps8, though it is well possible that the entire CORVET complex is required for this pathway, and not just Vps3 and Vps8. As such, an exclusive Vps3-Vps8 function, alone or together with VIPAS39/Vps33B requires further evidence. Along this line, the biochemical evidence for such a complex is rather circumstantial. I do not see any evidence for a transient complex, which the authors postulate in their study and embed in a rather speculative model. Thus, their model remains unproven by the data provided in this study. A main difficulty is the proposed direct interaction of Vps3 and Vps8,

which is based exclusively on a single pull-down. Considering that these proteins normally function as part of a hexameric complex, such far-fetched assumptions require further proof. In the absence of further evidence, the entire model of a Vps3-Vps8 complex and its crosstalk with the VIPAS39/Vps33B complex remains purely speculative.

Specific issues that need to be addressed:

1.) In Figure 1a the authors show the localization of Vps8 and Vps3 to the same structures, which appear to be negative for endocytosed BSA-Au. In the left panel the Vps3 construct is described as GFP-Vps3, whereas in the right panel a Vps3-GFP construct is used. Is this the same construct in both experiments?

2.) In order to support the co-localization data from Figure 1, a GFP-Pulldown using different CORVET and HOPS subunits was performed, showing a strong affiliation between Vps3 and Vps8. However, this remains to be the only data set supporting an interaction between Vps3 and Vps8 apart from the not so unexpected colocalization. This point requires further experimentation. The authors should use Vps8 as a bait in their pull-down, or better use purified proteins to map out the interaction site between Vps3 and Vps8. It should be noted that the Plemel et al. study, which focused on potential subcomplexes of CORVET, did not identify a specific interaction site, and there is also no in vivo proof of a functional role of such a subcomplex in yeast. Such a complex should also have a different molecular weight compared to the entire CORVET complex, which could be tested with complexes from cells.

3.) The authors performed co-immunoprecipitations (IP) to show an interaction between Vps3 and Vps8 with VIPAS39, Vps33B or both of these combined (Fig. 2e).

a. The blots of the GFP-pulldowns (Vps3 and Vps11) show a different picture regarding the control where only VIPAS39-mCherry is added (lane 4). Whereas no band can be seen on the blot for GFP-Vps3, a strong signal is observed in the experiment using GFP-Vps11. The authors do not offer any explanation for these discrepancies.

b. Apparently different sample volumes of the total cell lysates were used, when comparing the last two lanes of the GFP-Vps3 pulldown experiment, with the control lanes 3 & 5. The amount of Vps33B detected on the blot varies significantly and thus may influence any results gained from this experiment. The authors need to repeat this, using the same amount or preparation of TLC as an input. Otherwise the statement cannot be made.

c. The authors propose an interaction between Vps3 and Vps8 with the VIPAS39/Vps33B complex, although no Vps33B can be detected in the Co-IP experiments. There is no explanation given regarding this problem. It is also rather confusing that the complex of the two proteins

4.) The authors show a genetic and molecular interplay between Vps33B and the SNARE protein VAMP3. While I agree with their data on the specificity of the interaction between Vps33B and VAMP3 (Fig. 3c), it is confusing that they are not able to pull VIPAS39-mCherry along with Vps33B in their VAMP3-GFP pulldown (Fig. 3b). Why is the interaction not observed between VIPAS39 and Vps33b in any of their assays if it has been observed by others? In this regard, their model (Figure 3f) remains unproven by their own experiments.

5.) In their integrin assays, the authors are able to show a deficit in recycling of integrins (Fig. 4), as well as the consequences of this phenotype regarding adhesion and cell motility (Fig. 5). However, these experiments lack several controls.

a. In Figure 4b and d, the authors need to test single knock-downs of Vps3 and Vps8, since the function of the CORVET complex is abolished in these cells. This is a general problem in all of the following data sets, which the authors do not comment on at all. Furthermore, additional controls

featuring knock-downs of Vps11, Vps18, Rab11 and VIPAS39 are important additional controls that should be included. A similar phenotype would agree with a function of all proteins within this pathway.

b. A knock-down of VIPAS39 would be a more elegant control than Vps33B. This would abolish any proposed interaction between Vps33B and VAMP3, as well as the interaction with Vps3 and Vps8, which the authors postulate. It would also not disrupt the early endosomal fusion machinery.

6.) In Figure 6 the authors propose a model which is not supported by the presented data. There is no evidence for a disassembly of CORVET, a dimeric Vps3-Vps8 complex and tetrameric complex with VIPAS39 and Vps33B given in their data set. Without a demonstration that such intermediates exist it remains completely speculative.

Reviewer #3, expert in integrin trafficking (Remarks to the Author):

In this novel and interesting manuscript, Jonker et al. show that cargoes are transported from early endosomes to recycling endosomes by a multiprotein complex involving Vps3, Vps8, VIPAS39, Vps33B, and VAMP-3. This is fascinating because the machinery that couple early endosomes has not previously been described, hence answering a long standing question in the field. Furthermore, Vps3 and Vps8 have previously been described as components of the CORVET complex which controls early endosomal fusion events, and it is interesting that the authors now reveal an apparently CORVET-independent function for these. The authors provide evidence that this newly identified functional complex is required for trafficking of integrins, but not transferrin receptors, and show that Vps3/8 and Vps33B are required for integrin function in cell spreading and cell migration. I think that this is a really interesting addition to the field, which will appeal to the trafficking and adhesion communities. However, at present I have a few reservations, mainly around the argument that this is indeed a multiprotein complex that functions as the authors suggest in their final model.

Major comments:

1. I find the use of the word 'interact' confusing in this manuscript. Interact implies that two proteins directly bind to each other, but this is not clearly shown in the manuscript (co-IPs aren't sufficient evidence). I think it is better to say 'associate', or show direct interaction using recombinant protein in vitro.
2. Important controls showing that effects of knockdown are not off target (e.g. rescue experiments) are lacking throughout. This should be performed for some key experiments with Vps3/8/33B (and VIPAS39, see comment 9) knockdown.
3. The interaction and functional interplay between Vps33B and VIPAS39 has previously been published, but is not well characterized/confirmed here. Do Vps33B and VIPAS39 indeed associate in this cell system? I think this is important as the authors show by co-IP that Vps3/8 bind to VIPAS39, and Vps33B interacts with VAMP-3, but not how these two sub-complexes might interact to bring about fusion. Could a less stringent buffer (or different technique e.g. FRET?) be used to show that a complex between Vps3/8, VIPAS39 and Vps33B does exist?
4. In figure 2e, it appears that Vps11 does significantly associate with VIPAS39 mCherry, although this is present in the -GFP-Vps11 control (lane 4). Is this real? Could a more representative blot (or exposure) be shown to support the author's conclusion that Vps11 and VIPAS39 do not form a complex?
5. In Figure 3a, co-localization of transferrin with dextran is used as a surrogate to assess transferrin recycling. However, co-localization at this time point is already very high, is it realistic to expect to see an increase? Perhaps this has been shown before, or the authors have evidence that interfering with recycling does this (Rab22 knockdown for example), but they should cite/show this, or use a more direct measure of transferrin recycling (radioactive/fluorescent transferrin assays have been published).
6. In figures 4 and 5, Vps3 and 8 are suppressed simultaneously by knockdown. Is it necessary to

knock both down, or can effects on integrin traffic/migration be seen by knockdown of each alone?

7. In figure 4, is the same level of internal integrin achieved before recycling is initiated? This is not clear.

8. Integrin alpha-5 beta-1 is a fibronectin receptor, so it makes sense that impairing the pathway influences adhesion/migration on fibronectin. However, adhesion/migration on collagen is also decreased. Does the pathway handle other integrins, e.g. collagen binding (e.g. alpha-2), or for that matter other fibronectin binding integrins (alpha-v beta-3)?

9. The effect of VIPAS39 knockdown on integrin traffic and adhesion/migration is not investigated. This is important to show that the proposed link between Vps3/8 and Vps33B is itself critical.

Minor comments:

1. In the introduction on page 2 line 49-50 the authors imply that Rab11 'slow' recycling requires Rab4- is this what they mean? I thought these were independent of each other?

2. p16 ref 39 invalid citation?

Reviewer #1:

In some cases, I would be more cautious with the interpretation of the structures shown. In thin sections, vesicles, tubules and cisternae may appear with the same profiles. In particular, I would like to draw attention to the Golgi apparatus. Recycling endosomes may be located in close vicinity to Golgi apparatus stacks. In some of the images, such as in Fig.1d-top panel and Fig.3e-both panels, structures reminiscent of stacks/mini-stacks of Golgi apparatus cisternae are visible. In the legend to Fig.4f, G=Golgi is indicated but the respective description cannot be found in the picture. Since the spatial vicinity might be of functional significance, it would be of particular interest to show the relation of recycling endosomes to Golgi apparatus compartments.

It is indeed true that the distinction between membranes of the trans Golgi network (TGN) and recycling endosomes is not always easy to make, since these are both convoluted tubulo-vesicular membranes that are partially coated with clathrin. Moreover, as the reviewer indicates, recycling endosomes can sometimes be found within the TGN area. However, TGN and recycling vesicles can also be seen well apart, as illustrated in the picture below, which clearly shows that Vps3 and Vps8 are not present on TGN vesicles (red arrows), but on a cluster of vesicles at distance of the Golgi stack (white arrows).

In addition to distinguishing TGN and recycling endosomes by position, these membranes can be labeled with specific marker proteins. In our paper, we show that Vps3 and Vps8 co-localize with Rab4, which is an accepted marker for early and recycling endosomes and not present on the TGN. Taken together, we are confident in our definition that Vps3 and Vps8-positive vesicles are involved in recycling and not part of the TGN.

We corrected the legend of figure 4 and removed the indication G=Golgi.

Of note, Judith Klumperman has extensively studied the fine structure of TGN and endosomes (Klumperman J, Raposo G. *The complex ultrastructure of the endolysosomal system*, 2014 and Klumperman J. *Architecture of the mammalian Golgi*, 2011).

Immuno-electron microscopy showing Vps3 and Vps8-positive recycling vesicles to a nearby Golgi stack. Golgi vesicles (red arrows) are not labeled, whereas the typical recycling vesicles (white arrows) are positive for both Vps3 (15nm gold) and Vps8 (10nm gold). Bar, 200nm. EE = early endosome; LE = late endosome, M = Mitochondrion.

Reviewer #2:

1.) In Figure 1a the authors show the localization of Vps8 and Vps3 to the same structures, which appear to be negative for endocytosed BSA-Au. In the left panel the Vps3 construct is described as GFP-Vps3, whereas in the right panel a Vps3-GFP construct is used. Is this the same construct in both experiments?

- Indeed, these are both are GFP-Vps3, we have corrected this.

2.) In order to support the co-localization data from Figure 1, a GFP-Pulldown using different CORVET and HOPS subunits was performed, showing a strong affiliation between Vps3 and Vps8. However, this remains to be the only data set supporting an interaction between Vps3 and Vps8 apart from the not so unexpected colocalization. This point requires further experimentation. The authors should use Vps8 as a bait in their pull-down, or better use purified proteins to map out the interaction site between Vps3 and Vps8. It should be noted that the Plemel *et al.* study, which focused on potential subcomplexes of CORVET, did not identify a specific interaction site, and there is also no *in vivo* proof of a functional role of such a subcomplex in yeast. Such a complex should also have a different molecular weight compared to the entire CORVET complex, which could be tested with complexes from cells.

- As the reviewer suggested, we performed GST-pulldown assays with purified proteins. This shows that Vps3 binds directly to Vps8 (new fig. 2b). Moreover, using this assay we could pinpoint the Vps8 binding site of Vps3 to the C-terminus (new fig. 2b).
- Indeed, the study of Plemel *et al.* did not identify a Vps3/8 subcomplex in yeast. However, we suggest that the Vps3/8 complex is involved in integrin recycling in the same pathway as the CHEVI complex, which is also not found in yeast (line 247, discussion). Therefore, we speculate that like the CHEVI complex, the Vps3/8 complex arose evolutionary after the CORVET complex to adapt to processes like cell matrix attachment in higher eukaryotes.

3.) The authors performed co-immunoprecipitations (IP) to show an interaction between Vps3 and Vps8 with VIPAR39, Vps33B or both of these combined (Fig. 2e).

a. The blots of the GFP-pulldowns (Vps3 and Vps11) show a different picture regarding the control where only VIPAS39-mCherry is added (lane 4). Whereas no band can be seen on the blot for GFP-Vps3, a strong signal is observed in the experiment using GFP-Vps11. The authors do not offer any explanation for these discrepancies.

- We are very grateful for this comment, since following up on this question we discovered that the mCherry-VIPAS39 construct binds to the anti-GFP antibody (11814460001 Roche) used for coating the beads for the pulldown studies. To investigate the validity of the proposed Vps3 – VIPAS39 binding we switched the tag on the VIPAS39 construct to HA, validated that this shows no background binding to the beads, and repeated the GFP pulldown. Using this set up, binding of Vps3 to VIPAS39 was not detectable in standard experimental conditions (new fig 3d), however, after prolonged exposure of the western blots we could still detect some interaction between Vps3 and VIPAS39, indicating that there may be a weak or transient interaction (new fig S2). Based on these new experiments we tuned down this message in the paper and now suggest that transient Vps3-CHEVI interactions may occur, rather than claiming that there is a stable complex between CHEVI and Vps3.

1	GFP	5	GFP-Vps11 + Vps33B-HA-V5-his
2	GFP-Vps11	6	GFP-Vps11 + mCherry-VIPAS39
3	Vps33B-HA-V5-his	7	GFP-Vps11 + Vps33B-HA-V5-his + mCherry-VIPAS39
4	mCherry-VIPAS39	8	Vps33B-HA-V5-his + mCherry-VIPAS39

b. Apparently different sample volumes of the total cell lysates were used, when comparing the last two lanes of the GFP-Vps3 pull-down experiment, with the control lanes 3 & 5. The amount of Vps33B detected on the blot varies significantly and thus may influence any results gained from this experiment. The authors need to repeat this, using the same amount or preparation of TLC as an input. Otherwise the statement cannot be made.

- We indeed had unequal expression levels in our original samples. We addressed this by repeating the experiment using a different transfection reagent, i.e. Effectene (Qiagen instead of polyethylamine (PEI)-based. This resulted in consistent and reproducible co-expression levels of the proteins. In addition we now include the loading controls in the blots (new fig. 3d).

c. The authors propose an interaction between Vps3 and Vps8 with the VIPAS39/Vps33B complex, although no Vps33B can be detected in the Co-IP experiments. There is no explanation given regarding this problem. It is also rather confusing that the complex of the two proteins

- In our new pull-down experiments we clearly see an interaction between GFP-Vps33B and HA-VIPAS39 (fig. 3d, lane 7 in both panels). With respect to an interaction of VIPAS39 with Vps3, see our answer to point 3a.

4.) The authors show a genetic and molecular interplay between Vps33B and the SNARE protein VAMP3. While I agree with their data on the specificity of the interaction between Vps33B and VAMP3 (Fig. 3c), it is confusing that they are not able to pull VIPAS39-mCherry along with Vps33B in their VAMP3-GFP pull-down (Fig. 3b). Why is the interaction not observed between VIPAS39 and Vps33b in any of their assays if it has been observed by others? In this regard, their model (Figure 3f) remains unproven by their own experiments.

- In our new pull-down experiments, we clearly see an interaction between GFP-Vps33B and HA-VIPAS39 (fig. 3d, lane 7 in both panels).
- Regarding the model (old figure 3f), our new data indicate that interactions between Vps3 and the CHEVI complex are weak or transient (point 3a). We therefore tuned down our conclusions on the Vps3/8 – CHEVI complex interactions. Consequently, we also removed data on the CHEVI complex and VAMP3 (old figures 3a, 3b, 3c, 3d and 3f), which in our opinion are now outside the focus of this paper.

5.) In their integrin assays, the authors are able to show a deficit in recycling of integrins (Fig. 4), as well as the consequences of this phenotype regarding adhesion and cell motility (Fig. 5). However, these experiments lack several controls.

a. In Figure 4b and d, the authors need to test single knock-downs of Vps3 and Vps8, since the function of the CORVET complex is abolished in these cells. This is a general problem in all of the following data sets, which the authors do not comment on at all. Furthermore, additional controls featuring knock-downs of Vps11, Vps18, Rab11 and VIPAS39 are important additional controls that

should be included. A similar phenotype would agree with a function of all proteins within this pathway.

- We studied the effect of single knockdowns of Vps3 and Vps8 in the adhesion assay (Suppl fig 4), which is a straightforward and representative assay for integrin function. This showed that single knockdown of either Vps3 or Vps8 reduces cell adhesion, albeit less stringent than the combined knockdown (fig S4a). We interpret these data that residual expression in single knockdown cells (knockdown efficiency is 80-90%), allows formation of a small amount of complex, while in the double knockdown cells complex formation is more completely abrogated (fig. S4a).
- The request to include knockdowns of core subunits is a difficult one, since these core subunits are present in both HOPS and CORVET complexes. Knocking down core subunits therefore affects CORVET as well as HOPS functions, meaning that possible effects on integrin recycling could be the result of various trafficking defects. Furthermore, we observed significantly higher cell death when knocking down the core subunits Vps11 or Vps18 than after knockdown of Vps3 and/or Vps8. This hampered the execution of functional assays, as well as the more complex integrin recycling assays with these cells. Therefore, instead of proceeding with these knockdowns we performed proximity ligation assays (PLA) and additional immuno-EM to improve the evidence on the existence of a CORVET-independent Vps3/8 complex (new fig. 1b, 2c). The PLA assays show more signal between Vps3 and Vps8 than between Vps3 and the CORVET core subunits Vps11 or Vps16 (new fig. 2c) and the immuno-EM (new fig. 1b) shows co-localization of the CORVET core subunit Vps18 with Vps3 on early endosomes, whereas Vps18 is not found on the characteristic Vps3-positive recycling vesicles.

b. A knock-down of VIPAS39 would be a more elegant control than Vps33B. This would abolish any proposed interaction between Vps33B and VAMP3, as well as the interaction with Vps3 and Vps8, which the authors postulate. It would also not disrupt the early endosomal fusion machinery.

Because of the reasons stated above (see our answer to point 4), we removed the CHEVI complex data from the integrin recycling assays.

6.) In Figure 6 the authors propose a model which is not supported by the presented data. There is no evidence for a disassembly of CORVET, a dimeric Vps3-Vps8 complex and tetrameric complex with VIPAS39 and Vps33B given in their data set. Without a demonstration that such intermediates exist it remains completely speculative.

- We have altered this figure to accurately summarize the data presented in the revised manuscript (new fig. 6).

Reviewer #3:

1. I find the use of the word 'interact' confusing in this manuscript. Interact implies that two proteins directly bind to each other, but this is not clearly shown in the manuscript (co-IPs aren't sufficient evidence). I think it is better to say 'associate', or show direct interaction using recombinant protein in vitro.

- We have now performed a GST-pulldown that shows that Vps3 binds directly to Vps8 (new fig. 2b). Moreover, we have pinpointed the binding site to the C-terminus of Vps3. Finally, we performed proximity ligation assays (PLA) that show more signal between Vps3 and Vps8 than between Vps3 and the CORVET core subunits Vps11 or Vps16 (new fig. 2c). We find that with these additional data we can safely state that Vps3 and Vps8 interact with each other.

2. Important controls showing that effects of knockdown are not off target (e.g. rescue experiments) are lacking throughout. This should be performed for some key experiments with Vps3/8/33B (and VIPAS39, see comment 9) knockdown.

- To address this issue, we made the rescue constructs and performed rescue experiments in the integrin recycling assays and cell adhesion assays. However, we found that the transfection of the rescue constructs, in combination with the transfection of the siRNA oligo's, was too straining for the cells, which precluded the performance of reliable experiments. As an alternative approach to address the issue of off-target effects, we used two different single siRNA oligo's against Vps3 and two different oligo's against Vps8. As demonstrated in new fig. S4b, all four combinations reduced cell adhesion, which we took as representative assay for integrin function. These data indicate that reported effects of Vps3 and/or Vps8 depletion are specific and not due to an off-target effect.

3. The interaction and functional interplay between Vps33B and VIPAS39 has previously been published, but is not well characterized/confirmed here. Do Vps33B and VIPAS39 indeed associate in this cell system? I think this is important as the authors show by co-IP that Vps3/8 bind to VIPAS39, and Vps33B interacts with VAMP-3, but not how these two sub-complexes might interact to bring about fusion. Could a less stringent buffer (or different technique e.g. FRET?) be used to show that a complex between Vps3/8, VIPAS39 and Vps33B does exist?

- We optimized the expression conditions and used different constructs for reasons explained in comment 4 below. With this new co-IP protocol, we could show the interaction between Vps33B and VIPAS39, thus confirming that the CHEVI complex indeed associates in our cell system (new fig. 3d).

4. In figure 2e, it appears that Vps11 does significantly associate with VIPAS39 mCherry, although this is present in the -GFP-Vps11 control (lane 4). Is this real? Could a more representative blot (or exposure) be shown to support the author's conclusion that Vps11 and VIPAS39 do not form a complex?

We are very grateful for this comment, since following up on this question we discovered that the mCherry-VIPAS39 construct binds to the anti-GFP antibody (11814460001 Roche) used for coating the beads for the pulldown studies. To investigate the validity of the proposed Vps3 – VIPAS39 binding we switched the tag on the VIPAS39 construct to HA, validated that this shows no background binding to the beads, and repeated the GFP pulldown. Using this set up, binding of Vps3 to VIPAS39 was not detectable in standard experimental conditions (new fig 3d), however, after prolonged exposure of the western blots we could still detect some interaction between Vps3 and VIPAS39, indicating that there may be a weak or transient interaction (new fig S2). Based on these experiments we tuned down this message in the paper and now suggest that transient Vps3-CHEVI interactions may occur, rather than claiming that there is a stable complex between CHEVI and Vps3.

1	GFP	5	GFP-Vps11 + Vps33B-HA-V5-his
2	GFP-Vps11	6	GFP-Vps11 + mCherry-VIPAS39
3	Vps33B-HA-V5-his	7	GFP-Vps11 + Vps33B-HA-V5-his + mCherry-VIPAS39
4	mCherry-VIPAS39	8	Vps33B-HA-V5-his + mCherry-VIPAS39

5. In Figure 3a, co-localization of transferrin with dextran is used as a surrogate to assess transferrin recycling. However, co-localization at this time point is already very high, is it realistic to expect to see an increase? Perhaps this has been shown before, or the authors have evidence that interfering with recycling does this (Rab22 knockdown for example), but they should cite/show this, or use a more direct measure of transferrin recycling (radioactive/fluorescent transferrin assays have been published).

- The assay we used is an established assay (Ren M et al., 1998). In this paper the authors showed a clear increase in the co-localization between EEA1 and transferrin after inhibiting exit from early endosomes using a temperature block. Hence, despite the significant overlap found in control cells, this assay has a proven sensitivity to monitor changes in transferrin recycling. In addition, in a previous study Perini et al. 2014 also showed that Vps3 knockdown does not inhibit transferrin recycling, which supports our findings in new figure 4a and 4b.

6. In figures 4 and 5, Vps3 and 8 are suppressed simultaneously by knockdown. Is it necessary to knock both down, or can effects on integrin traffic/migration be seen by knockdown of each alone?

- We studied the effect of single knockdowns of Vps3 and Vps8 in the adhesion assay (Suppl fig 4), which is a straightforward and representative assay for integrin function. This showed that single knockdown of either Vps3 or Vps8 reduces cell adhesion, albeit less stringent than the combined knockdown (fig S4a). We interpret these data that residual expression in single knockdown cells (knockdown efficiency is 80-90%), allows formation of a small amount of complex, while in the double knockdown cells complex formation is more completely abrogated (fig. S4a).

7. In figure 4, is the same level of internalized integrins before recycling is initiated? This is not clear.

- Yes, we have similar amounts of internalized integrins before recycling is initiated. This is determined in the ELISA-based recycling assays (fig. 4f). For each experiment we determined the ratio of internalization between knockdown conditions and scrambled conditions. A ratio of 1 indicates that there is no difference in internalization. For all experiments we found an average internalization ratio of 1 with only a limited spread. We added these valuable data to the new manuscript (new fig. 4g).

8. Integrin alpha-5 beta-1 is a fibronectin receptor, so it makes sense that impairing the pathway influences adhesion/migration on fibronectin. However, adhesion/migration on collagen is also decreased. Does the pathway handle other integrins, e.g. collagen binding (e.g. alpha-2), or for that matter other fibronectin binding integrins (alpha-v beta-3)?

- We are indeed eager to know additional cargo for this pathway. We have tried identifying other integrins as cargos by Immuno-EM and by ELISA. These are our preliminary findings:
- The ELISA assay can to our knowledge only be performed using 2 established antibodies against human integrin subunits; $\alpha 5$ (#555651 from BD Biosciences) and $\beta 3$ (#555752 from BD Biosciences). We have performed the assay using the $\beta 3$ antibody but the expression of this integrin on HeLa cells is too low to reliably detect any changes in recycling. We have also tested several anti- αV antibodies but these did not generate sufficient signal in the ELISA assay.

- Similarly, only few anti-integrin antibodies work in immuno-EM. We have tried several antibodies against $\beta 3$ and αV , but these did not show any specific label. Therefore, we cannot indicate yet if or which other integrins are cargoes for this pathway.
- In our antibody-based recycling assays (fig 4c and fig S3), we use an anti- $\beta 1$ integrin antibody. These experiments suggest that Vps3/8 handles $\beta 1$ integrins in general. Indeed, as we detect differences in adhesion, spreading and migration on both FN and Col-I, we expect that this pathway does not exclusively regulate the recycling of $\alpha 5\beta 1$ but at least also of collagen-recognizing ($\beta 1$) integrin heterodimers. A further in-depth study of the cargo of this pathway falls outside the scope of the present paper and will be addressed in future studies.

9. The effect of VIPAS39 knockdown on integrin traffic and adhesion/migration is not investigated. This is important to show that the proposed link between Vps3/8 and Vps33B is itself critical.

- Our new data indicate that interactions between Vps3 and the CHEVI complex are weak or transient (point 4). We therefore tuned down our conclusions on the Vps3/8 – CHEVI complex interactions, and the here suggested role of VIPAS39 on integrin traffic. Consequently, we also removed data on the CHEVI complex and VAMP3 (old figures 3a, 3b, 3c, 3d and 3f), which in our opinion are now outside the focus of this paper.

Minor comments:

1. In the introduction on page 2 line 49-50 the authors imply that Rab11 'slow' recycling requires Rab4- is this what they mean? I thought these were independent of each other?

- The reviewer is correct, Rab4 is involved in the slow recycling pathway, but not always required y. We therefore changed "required" in the new manuscript to "involved" (new manuscript line 50).

2. p16 ref 39 invalid citation?

- This has been corrected.

REVIEWERS' COMMENTS:

Reviewer #1 (Remarks to the Author):

The main questions concerning the EM-part of this manuscript are convincingly answered by the authors.

Reviewer #2 (Remarks to the Author):

The authors provide a strongly revised version with several controls that are now included.

While I agree with their findings, I would strongly caution a couple of points that refer to the central message of their paper. The authors claim that their Vps3-Vps8 complex does not contain other subunits. This rests on three assays – the pull-down Figure 2A, the interaction of in vitro purified GST-Vps3 with Vps8 (Figure 2b) and the PLA signals.

In all of these assays, the authors overexpress their proteins, which are in other cells part of a large complex. Within the CORVET and HOPS complex, the C-terminal parts of these subunits (Vps3,8,11,16,18) seem to interact as shown for human and yeast HOPS and CORVET (Plemel et al., 2011; Guo et al., 2013). Consequently, they will have likely hydrophobic patches which are exposed in isolation and may also non-specifically interact. I therefore am doubtful that the two-fold more binding to Vps3 is significant compared to the other complex subunits (part a). Likewise, the pull-down in part b could be explained this way. With respect to the PLA, the same problems may occur as it again relies on overexpression.

I find this insufficient evidence to claim that Vps3 and Vps8 form a complex independent of the CORVET core. For this, they need to purify the endogenous complex and show convincingly that they find two populations, one with other core subunits, and one without.

I overall do think that this study has some very valid and important messages, including the consequences of the loss of the Vps3 and Vps8 and the localization relative to the Vps33B complex. My recommendation is that they strongly tune-down their claim of an independent Vps3-Vps8 complex both in the abstract and in the main text (as stated) – unless they provide this missing characterization. I am aware that this takes away a punch-line for now, though is in my view a much better decision than claiming a complex that is not properly characterized. I should emphasize that I do not exclude the possibility of such a complex or independent functions of Vps3 and Vps8, yet the authors do not provide the controlled data to make this statement.

Reviewer #3 (Remarks to the Author):

The authors have addressed my comments and I am happy that this manuscript is now ready for publication.

Point-by-point response to referees

Reviewer #1 and #3 had no further comments

Reviewer #2 (Remarks to the Author):

The authors provide a strongly revised version with several controls that are now included. While I agree with their findings, I would strongly caution a couple of points that refer to the central message of their paper. The authors claim that their Vps3-Vps8 complex does not contain other subunits. This rests on three assays – the pull-down Figure 2A, the interaction of in vitro purified GST-Vps3 with Vps8 (Figure 2b) and the PLA signals.

We add to this list of arguments the EM localizations: Vps3 and Vps8 are in addition to early endosomes also seen on recycling vesicles whereas Vps18 is not (figure 1b). This is for all 3 proteins under similar over-expression conditions.

In all of these assays, the authors overexpress their proteins, which are in other cells part of a large complex. Within the CORVET and HOPS complex, the C-terminal parts of these subunits (Vps3,8,11,16,18) seem to interact as shown for human and yeast HOPS and CORVET (Plemel et al., 2011; Guo et al., 2013). Consequently, they will have likely hydrophobic patches which are exposed in isolation and may also non-specifically interact. I therefore am doubtful that the two-fold more binding to Vps3 is significant compared to the other complex subunits (part a). Likewise, the pull-down in part b could be explained this way. With respect to the PLA, the same problems may occur as it again relies on overexpression. I find this insufficient evidence to claim that Vps3 and Vps8 form a complex independent of the CORVET core. For this, they need to purify the endogenous complex and show convincingly that they find two populations, one with other core subunits, and one without.

The reviewer is right in that over-expression may cause artefacts and that these may be different for the different subunits tested here.

I overall do think that this study has some very valid and important messages, including the consequences of the loss of the Vps3 and Vps8 and the localization relative to the Vps33B complex. My recommendation is that they strongly tune-down their claim of an independent Vps3-Vps8 complex both in the abstract and in the main text (as stated) – unless they provide this missing characterization. I am aware that this takes away a punch-line for now, though is in my view a much better decision than claiming a complex that is not properly characterized. I should emphasize that I do not exclude the possibility of such a complex or independent functions of Vps3 and Vps8, yet the authors do not provide the controlled data to make this statement.

We previously performed a number of endogenous pulldowns (see van de Kant *et al.*, 2015) but ran into problems due to a lack of suitable antibodies for all subunits. We agree with the reviewer that characterizing the endogenous complexes would be the ultimate proof of our claim, but currently do not see how to achieve this. We therefore accept the reviewers' suggestion to tone down the message of an independent complex.